# Intervention reducing malaria parasite load in vector mosquitoes: No impact on *Plasmodium falciparum* extrinsic incubation period and the survival of *Anopheles gambiae*

Edwige Guissou[1,2,3,4]*, Dari Frédéric Da[1], Domombabele François de Sales Hien[1], Koudraogo Bienvenue Yameogo[1], Serge Rakiswende Yerbanga[1], Georges Anicet Ouédraogo[3], Kounbobr Roch Dabiré[1], Thierry Lefèvre[1,2], Anna Cohuet[2]*

1 Institut de Recherche en Sciences de la Santé, Bobo-Dioulasso, Burkina Faso, 2 MIVEGEC, Montpellier University, IRD, CNRS, Montpellier, France, 3 Université Nazi Boni, Bobo-Dioulasso, Burkina Faso, 4 Ecole Normale Supérieure, Koudougou, Burkina Faso

* edwigeguissou@yahoo.fr (EG); anna.cohuet@ird.fr (AC)

**Data Availability Statement:** The data and scripts are available on DataSuds repository: https://

## Abstract

In the fight against malaria, transmission blocking interventions (TBIs) such as transmission blocking vaccines or drugs, are promising approaches to complement conventional tools. They aim to prevent the infection of vectors and thereby reduce the subsequent exposure of a human population to infectious mosquitoes. The effectiveness of these approaches has been shown to depend on the initial intensity of infection in mosquitoes, often measured as the mean number of oocysts resulting from an infectious blood meal in absence of intervention. In mosquitoes exposed to a high intensity of infection, current TBI candidates are expected to be ineffective at completely blocking infection but will decrease parasite load and therefore, potentially also affect key parameters of vector transmission. The present study investigated the consequences of changes in oocyst intensity on subsequent parasite development and mosquito survival. To address this, we experimentally produced different intensities of infection for *Anopheles gambiae* females from Burkina Faso by diluting gametocytes from three natural *Plasmodium falciparum* local isolates and used a newly developed non-destructive method based on the exploitation of mosquito sugar feeding to track parasite and mosquito life history traits throughout sporogonic development. Our results indicate the extrinsic incubation period (EIP) of *P. falciparum* and mosquito survival did not vary with parasite density but differed significantly between parasite isolates with estimated $EIP_{50}$ of 16 (95% CI: 15–18), 14 (95% CI: 12–16) and 12 (95% CI: 12–13) days and median longevity of 25 (95% CI: 22–29), 15 (95% CI: 13–15) and 18 (95% CI: 17–19) days for the three isolates respectively. Our results here do not identify unintended consequences of the decrease of parasite loads in mosquitoes on the parasite incubation period or on mosquito survival, two key parameters of vectorial capacity, and hence support the use of transmission blocking strategies to control malaria.

dataverse.ird.fr/dataset.xhtml?persistentId=doi:10.23708/OJCZQD.

**Funding:** The work described in our manuscript was funded by support from the following sources: AC received support from the Malaria Vaccine Initiative, a program of the global non-profit PATH organization (Seattle, USA) and the European Union's Horizon 2020 research and innovation program under grant agreement No 733273. TL received support from the ANR Grant "STORM" No. 16-CE35-0007. DFDSH received support from the JEAI IRD program Grant No. AAP2018_JEAI_PALUNEC. The funders had no role in study design, data collection and analysis, decision to publish, or preparation of the manuscript.

**Competing interests:** The authors have declared that no competing interests exist.

## Author summary

In the fight against malaria, it is recognized that the use of several complementary strategies is necessary to significantly reduce transmission and improve human health. Among these, transmission blocking strategies such as transmission blocking vaccines or drugs, aim to block the development of the parasites within mosquito vectors. This approach should prevent infection in most mosquitoes feeding on infectious hosts and thus block transmission. However, in some cases it may only reduce parasite load without fully clearing the infection. Here we identified potential risks: if reducing parasite load would reduce the incubation period of the parasite in mosquitoes or increase the longevity of the mosquitoes, undesirable consequences may occur with an increased efficiency of these vectors to transmit infection to humans. We tested these hypotheses and experimentally produced different infection loads in *Anopheles gambiae* by using dilutions of *Plasmodium falciparum* isolates from naturally infected human donors. We observed that the longevity of mosquitoes and the incubation period of the parasites were not affected by the parasite load. This is not consistent with the unintended risks that we identified and thus strengthens the potential of transmission blocking interventions in the toolbox to combat malaria.

## Introduction

Despite significant progress in the fight against malaria in the last two decades, nearly half of the world's population remains at risk of contracting the disease. The African region is the most affected, accounting for 94% of the global malaria burden [1]. Malaria control mainly relies on the use of antimalarial drugs, with an important contribution from artemisinin-based combination therapies, and vector control with the use of long-lasting insecticidal nets and indoor residual spraying. These tools enabled a significant reduction in the incidence and mortality due to malaria since the beginning of the century, but this decline has worryingly stalled in some countries and has even been reversed in some others in recent years with the spread of drug-resistance among parasites [2,3] and insecticide resistance in the main mosquito vectors [4]. As a complement to conventional tools targeting parasites in humans or seeking to kill mosquito vectors, new tools targeting parasites within mosquitoes appear promising [5–7]. These approaches, known as transmission blocking interventions (TBIs), target parasites within the mosquitoes where they are less numerous and express less variability than in human hosts. The principle of the current TBI candidates is to administrate drugs [8,9] or vaccines [10–13] to the human population so that the mosquitoes will not only ingest infectious gametocytes but also the drug or antibodies when taking a blood meal. These blocking agents will impede the development of infection at early stage within vectors and thereby reduce the subsequent exposure of human populations to infectious mosquitoes. The efficiency of TBIs is often measured by comparing oocyst intensities between groups of mosquitoes exposed to an infectious blood meal with *versus* without the blocking agent, and it has been shown to depend on the intensity of infection in the control group of mosquitoes [14–16]. In other words, when the intensity of infection is moderate or low ($< 5$ oocysts per mosquito in the absence of a TBI, as often found in naturally infected mosquitoes [17–19]), transmission-blocking strategies will be much more effective in reducing the prevalence of infection in mosquitoes compared to situations where mosquitoes carry higher intensities of infection. However, in nature, it has been shown that the distribution of oocysts is highly overdispersed, with a significant proportion of infected mosquitoes carrying dozens of oocysts and where a few mosquitoes harbor very high oocyst densities ($> 50$ oocysts per mosquito) [20].

Therefore, in mosquitoes exposed to high densities of parasites it is expected that imperfect TBIs will reduce the number of oocysts, but will be ineffective at completely blocking infection, which may lead to unexpected consequences.

Pathogen density is an important factor contributing to the virulence and transmission of disease [21,22]. Consequently, interventions altering pathogen density deserve attention. In general, theoretical assumptions predict that reduced densities in populations are associated with competitive release and increased fitness [23]. However the patchy resources, the diversity of hosts and environmental conditions complicate the predicted consequences of varying pathogen density for the transmission of disease [24]. In malaria vectors, host-parasite interactions shape important parameters of transmission [25,26]. Among them, the duration of the parasite's development within the mosquito, from the ingestion of gametocytes to the invasion of salivary glands by sporozoites, also called the extrinsic incubation period (EIP), and mosquito longevity are the most influential [27]. These two parameters are critical as the EIP is often as long as a mosquito's average lifespan thus limiting the time window for sporozoite transmission before mosquito death [28]. This intimate relationship between parasite EIP and mosquito longevity should theoretically favour the rapid development of parasites to their transmission stages (i.e. sporozoites in salivary glands), but trade-offs between multiple traits in response to the mosquito environment may constrain this evolution [26,29,30]. Evidence for genetic and environmental variability of EIP exists, although remains scarce. Still, it is well described that EIP is affected by temperature, with warming temperatures, until a threshold, that speed up the parasite development [31–35]. There are also suggestions of interspecific genetic influence as some *Plasmodium* species develop faster than others: *P. mexicanum* transmitted by short-lived sandflies has a short EIP compared to other *Plasmodium* species [36]. Whether variation in malaria parasite density may also influence its developmental schedule in mosquitoes remains to be explored. Other parasite species, or blood-stage malaria parasites [37,38] have been shown to speed up their investment into transmission stages in conditions of stress, when their transmission is compromised by the potential death of the host. This suggests that damage caused by malaria parasites to the vector, if related to parasite density, may induce density-dependant consequences on EIP. Moreover, EIP is now known to be affected by the nutritional status of the mosquito host, in larval or adult stages [39–42]. Parasite density in the mosquito may then interact with limited nutritional resources, which should become less restrictive if the intensity of infection decreases, allowing faster development [43].

Closely related to EIP, but easier to study and more documented, is mosquito survival in respect to *Plasmodium* infection. The effect of *Plasmodium* infection on the survival of its vectors has long been disputed with conflicting observations [44,45] but a general trend appears in natural and artificial combinations of vectors and parasites for negative effects of infection on mosquito longevity in combination with other stresses, such as; hydric stress [46], resistance to insecticides [47], exposure to insecticides [48] poor nutritional resources [49,50] exposure to predators [51] and was also found dependent on parasite genetics [44,49]. However, a density-dependent effect of *Plasmodium* infection on vector survival was observed only in experimental vector–parasite combinations or avian malaria systems [24,52–54] and to our knowledge never for the most deadly human parasite, *Plasmodium falciparum*. It is therefore important to investigate the effect of parasite density on this key transmission parameter and investigate whether interventions to control *P. falciparum* may unintentionally increase the life expectancy of infected vectors by reducing the intensity of infection and thus possibly facilitating the successful and sustained transmission of the pathogen [55].

As interventions to control malaria may affect parasite intensity in mosquitoes and possibly affect key parameters of vectorial transmission, the present study investigated the

consequences of changes in the intensity of infection in vector mosquitoes on parasite development and mosquito survival. We experimentally produced different intensities of infection in *Anopheles gambiae* females by diluting gametocytes from natural isolates of *P. falciparum* and used a newly developed non-destructive method [56] based on the exploitation of mosquito sugar feeding to track parasite and mosquito life history traits throughout sporogonic development.

## Results

### Effect of infectious blood dilution on the prevalence and intensity of infection in mosquitoes

**Infection prevalence and intensity in mosquito gut at 7 days post blood meal (dpbm).** *An. gambiae* females were experimentally infected with the blood from one of three naturally infected gametocyte carriers (parasite isolates A, B and C) in Burkina Faso. The gametocyte-infected blood of each carrier was diluted to experimentally reduce the density of infectious gametocytes and create a range of parasite loads in mosquitoes. The midguts of 193 females were dissected at 7 dpbm for oocyst observation, among which 132 were positive for *P. falciparum* (68.4%). Dilution had no significant effect on the prevalence of infection (LRT $X^2_1$ = 1.8, P = 0.177, Fig 1a), but had the intended effect of strongly reducing the intensity of infection among oocyst-infected mosquitoes throughout the dilution range (LRT $X^2_1$ = 7.1, P = 0.008, Fig 1b).

**Infection prevalence and intensity in mosquito head/thoraces upon death.** At 7 dpbm, 269 *An. gambiae* females challenged with either parasite isolate A, B or C and that received the different dilution treatments (1/1, 1/3, 2/3 for parasite isolates A and B and 1/1, 1/3, 1/8 for parasite isolate C), were placed in individual tubes for saliva collection on cotton balls soaked with a 10% glucose solution. Upon mosquito death, the amount of parasite DNA in the heads/thoraxes of the females used to collect saliva was assessed using qPCR. Of a total of 269 *An. gambiae* females placed in individual tubes, 201 (75%) were found positive for *P. falciparum* by qPCR of the heads/thoraxes of mosquitoes at their time of death. The proportion of sporozoite-infected mosquitoes significantly increased with the density of infectious gametocytes (LRT $X^2_1$ = 4.7, P = 0.030, Fig 1c). The global percentage of positive heads/thoraxes reached 88% (151 out of 172) when females that died before 14 dpbm (the time generally considered for sporozoites to have invaded mosquito salivary glands) were excluded. Consistent with observations made of mosquito midguts, the amount of parasite DNA in the heads and thoraxes of infected mosquitoes showed a positive relationship with gametocytemia (LRT $X^2_1$ = 24.3, P< 0.001, Fig 1d).

### Effect of parasite density on EIP

The presence of parasite DNA in the cotton balls used to collect saliva from infected mosquitoes (n = 201) was examined. A total of 1 997 cotton balls were analyzed and individual EIP was defined as the time between the infectious blood meal and the first day of positive qPCR detection of *P. falciparum* from a cotton wool substrate for a given infected female. Of the 201 females with an infected head/thorax, 102 (50.7%) generated at least one cotton ball containing detectable traces of parasite DNA. The infected females that did not produce any positive cottons over their lifespan were excluded from the analysis because no EIP values can be derived from these samples. The first positive cottons occurred at 9 dpbm in all dilution treatments. Among the mosquitoes that already produced a positive cotton ball, the proportion of positive cotton balls was 33, 81% (896 positive cottons out of 2650).

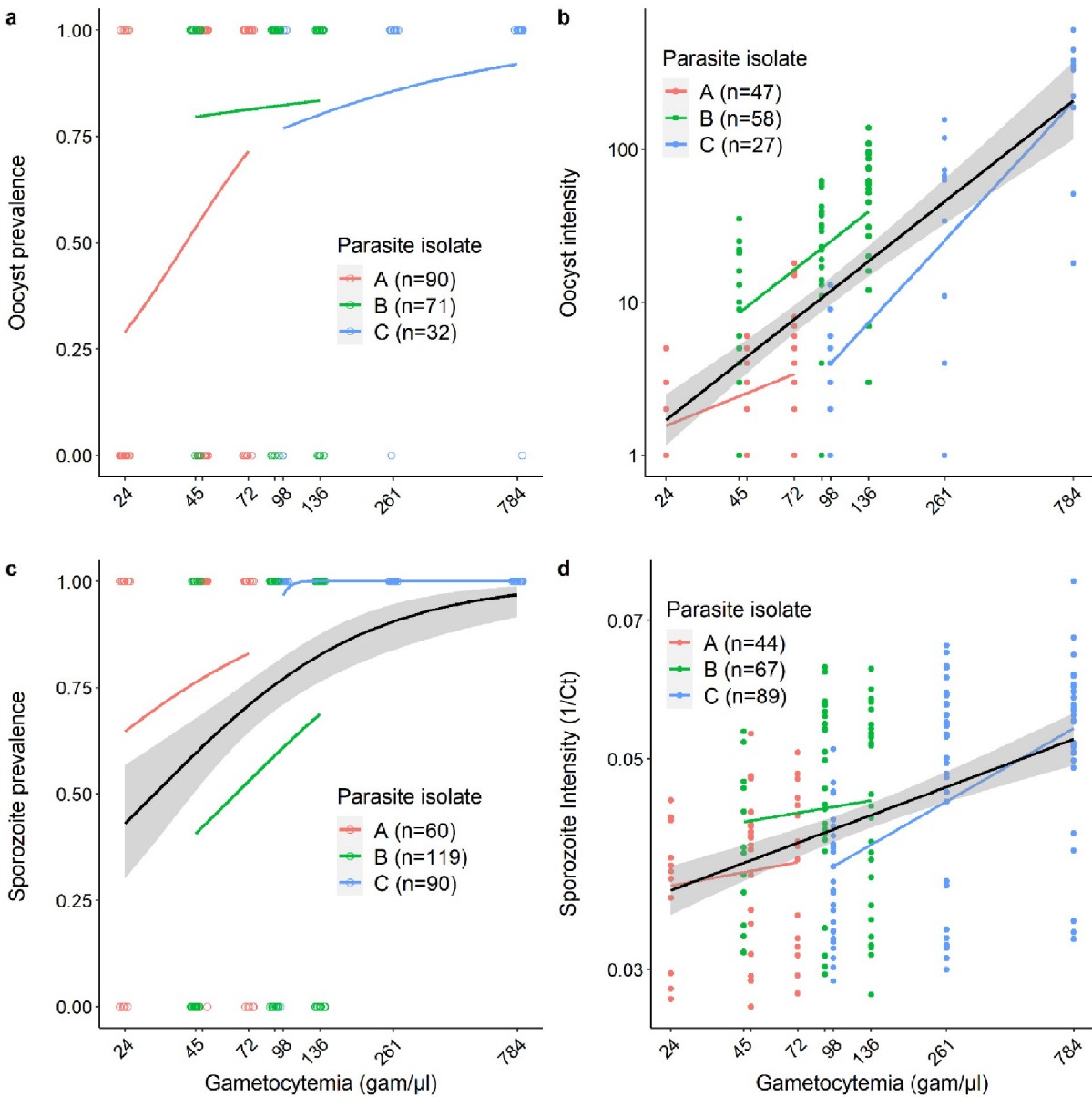

**Fig 1. Effect of infectious blood dilution on the prevalence and intensity of infection in mosquitoes.** (a) Oocyst prevalence 7 dpbm (number of mosquitoes with at least one oocyst in their midguts out of the total number of mosquitoes dissected) as a function of gametocytemia (the estimated number of infectious gametocytes per microliter of blood in each of the dilution groups: 24, 45, 48, 72, 91, 98, 136, 261 and 784 gametocytes/µl of blood, note that to avoid overlapping of the x-axis labels, the concentrations of 48 and 91 gam/µl are not indicated). Each colored circle represents a dissected mosquito (red: parasite isolate A with an initial gametocytemia of 72, green: parasite isolate B with initial gametocytemia of 136, and blue: parasite isolate C with an initial gametocytemia of 784). The colored lines represent the best-fit logistic growth curves for each parasite isolate. Note that the x-axis is on a log10 scale. (b) Oocyst intensity 7 dpbm (number of oocysts in infected mosquitoes) as a function of gametocytemia. Each colored circle represents a *P. falciparum* oocyst-positive midgut. The colored lines represent the linear relationship for each parasite isolate, while the black line (± se) for all data regardless of isolate origin. Note that the x- and y- axes are on a log10 scale. (c) Sporozoite prevalence (number of mosquitoes with heads/thoraxes detected positive to *P. falciparum* by qPCR at the time of death of the individual out of the total number of tested heads/thoraxes) as a function of gametocytemia. Each colored circle represents a tested head/thorax. The x-axis is on a log10 scale. (d) Amount of parasite DNA in mosquito heads/thoraxes expressed as the inverse of the qPCR cycle threshold (1/Ct, the higher the inverse of threshold cycle, the higher the intensity of infection). For each mosquito, 1/Ct value is the average over 4 to 6 technical replicates. The x- and y-axes are on a log10 scale. Each colored circle represents a *P. falciparum* positive head/thorax using qPCR.

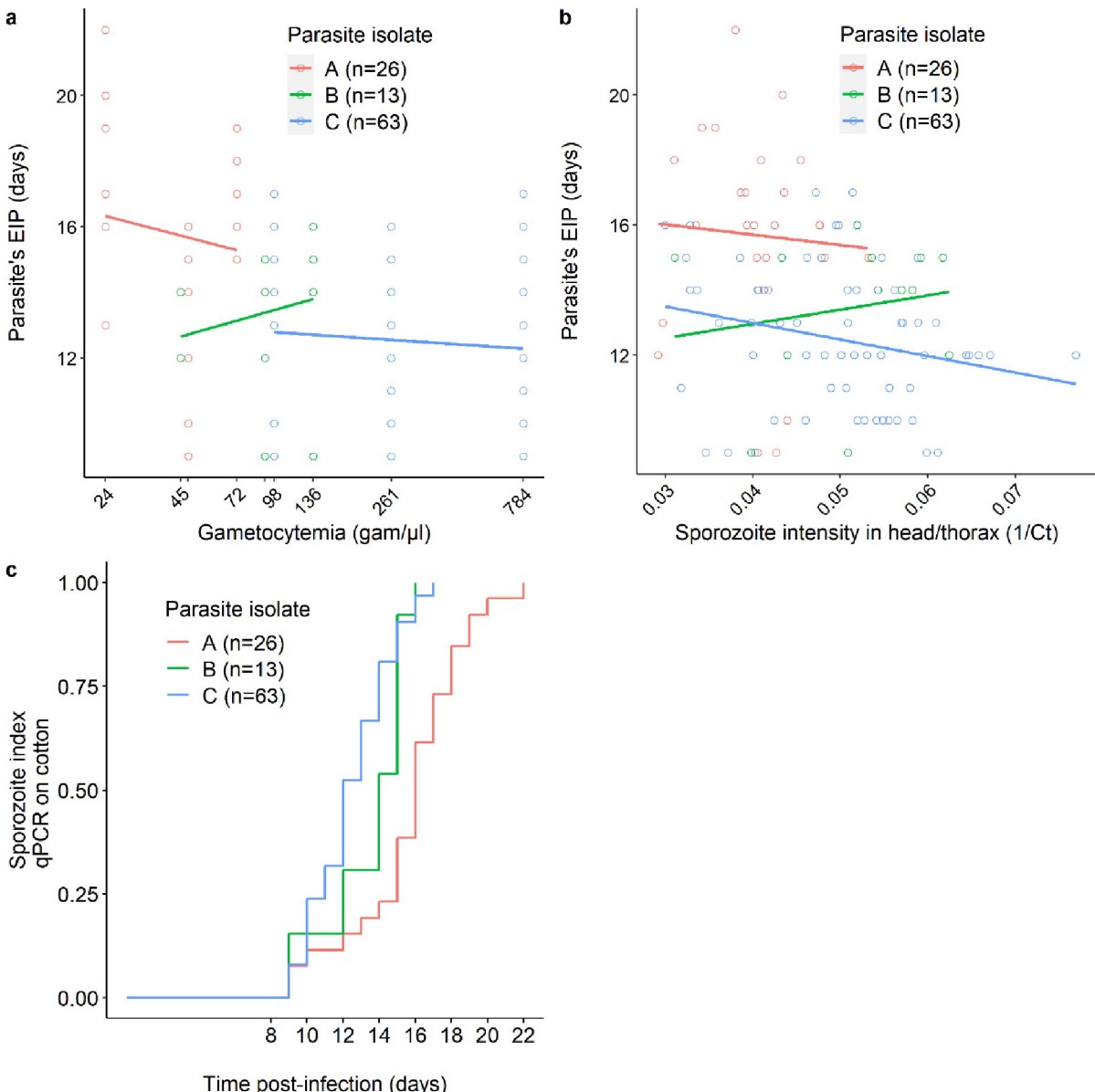

**Fig 2. Relationship between parasite density and EIP.** (a) EIP (the time between the infectious blood meal and the detection of *P. falciparum* in mosquito saliva collected from cotton balls) as a function of gametocytemia (the number of gametocytes per microliter of blood in each of the dilution groups: 24, 45, 48, 72, 91, 98, 136, 261 and 784 gametocytes/μl of blood, note that to avoid overlapping of the x-axis labels, the concentrations of 48 and 91 gam/μl are not indicated). Each colored circle represents an infected mosquito from which EIP was measured (red: parasite isolate A with an initial gametocytemia of 72, green: parasite isolate B with initial gametocytemia of 136, and blue: parasite isolate C with an initial gametocytemia of 784). The x-axis is on a log10 scale. (b) EIP as a function of 1/Ct in mosquito infected heads/thoraxes extracts (the higher the 1/Ct value, the higher the infection intensity). For each cotton ball, 1/Ct value is the average over 3 technical replicates. The colored lines in panels (a) and (b) represent the linear relationship for each parasite isolate. (c) Kaplan–Meier curves representing the temporal dynamics of sporozoite detection in cotton balls used to collect saliva from individual mosquitoes fed on each parasite isolate.

There was no effect of gametocytemia on EIP ($LRT\ X^2{}_1 = 0.9$, P = 0.341, Fig 2a). Similar results were obtained when the explanatory variable, the gametocytemia, was substituted by the actual intensity of infection found in the head/thorax of each individual female upon their death ($LRT\ X^2{}_1 = 1.0$, P = 0.317, Fig 2b). However, there was a significant effect of parasite

isolate ($LRT$ $X^2_2$ = 32.8, P < 0.001) with an estimated $EIP_{50}$ of 16 (95% CI: 15–18), 14 (95% CI: 12–16) and 12 (95% CI: 12–13) days for isolate A, B and C, respectively (Fig 2c).

## Effect of parasite density on mosquito survival

Mosquito survival was monitored daily for the 269 females placed in individual tubes, including 201 infected with *P. falciparum* and 68 fed on infectious blood but that did not develop an infection. Infected females survived better than those that did not become infected ($LRT$ $X^2_1$ = 14.2, P < 0.001, Fig 3a). No interaction between infection status and gametocytemia on mosquito survival was found ($LRT$ $X^2_1$ = 0, P = 0.968, Fig 3b). There was no effect of gametocytemia on the lifespan of infected mosquitoes ($LRT$ $X^2_1$ = 0.1, P = 0.783, Fig 3c) and no effect of infection intensity in the head/thorax of mosquitoes on the lifespan of infected mosquitoes ($LRT$ $X^2_1$ = 1.9, P = 0.164, Fig 3d). Finally, the lifespan of infected mosquitoes varied strongly depending on parasite isolate ($LRT$ $X^2_1$ = 46.6, P <0.001, Fig 3e), with a median longevity of 25 (95% CI: 22–29), 15 (95% CI: 13–15) and 18 (95% CI: 17–19) days for infected mosquitoes fed on isolate A, B and C, respectively.

## Discussion

In the present study, we questioned the importance of infection load in malaria-infected mosquitoes. We investigated the relationship between *P. falciparum* gametocyte densities in infectious blood and subsequent transmission parameters in mosquitoes, including infection prevalence, infection intensity at oocyst and sporozoite stages, and more originally the time taken for mosquitoes to become infectious (the parasite's EIP) and their survival. EIP and mosquito survival are key parameters for transmission [26,29,57] and we explored the extent to which an intervention affecting the intensity of infection could affect them. Because the outcome of infection in mosquitoes depends on various parameters, such as, gametocyte maturity and sex ratio [58–61], genetics [44,62–64], parasite multiplicity of infection [61–65], as well as environmental conditions [51–66] we generated experimental ranges of parasite loads from infectious blood samples so that for a given parasite isolate, only the density of infectious gametocytes varied. To do this, we diluted field collected gametocyte-containing blood isolates by a volume of the same blood sample after it had been exposed to heat inactivation and we used a range of dilutions to expose mosquitoes in controlled conditions. This generated a wide range of infection loads, mimicking natural high infection loads in some mosquitoes [20] and a number of parasites reduced by an imperfect TBI in other individuals.

The range of *P. falciparum* infectious gametocyte densities generated by dilution resulted in proportional oocyst intensities of infection in *An. gambiae* mosquitoes. This relationship between gametocyte density and oocyst load in mosquitoes was previously demonstrated in a study using the same dilution protocol [67]. This dilution procedure also confirms the positive correlation between gametocyte density and oocyst load observed in the range of natural gametocyte densities, while reducing variance due to uncontrolled confounding factors occurring in natural conditions [58]. Here we observed similar relationships between gametocyte densities and sporozoite load in mosquito head/thorax extracts, consistent with the linear relation between oocyst number and sporozoite load in salivary glands [68,69]. Surprisingly, the sporozoite prevalence we observed by qPCR in head/thorax extracts of mosquitoes from 14 dpbm (88%) were higher than oocyst prevalence in midguts at 7 dpbm (68.4%). This suggests qPCR for sporozoite detection could, (i) be more sensitive than microscope detection of oocysts, (ii) may produce false positives, or (iii) that infected females could survive better than exposed but non-infected females. These three hypotheses are not mutually exclusive; the fact that we observed an important proportion of females positive for *P. falciparum* in their heads

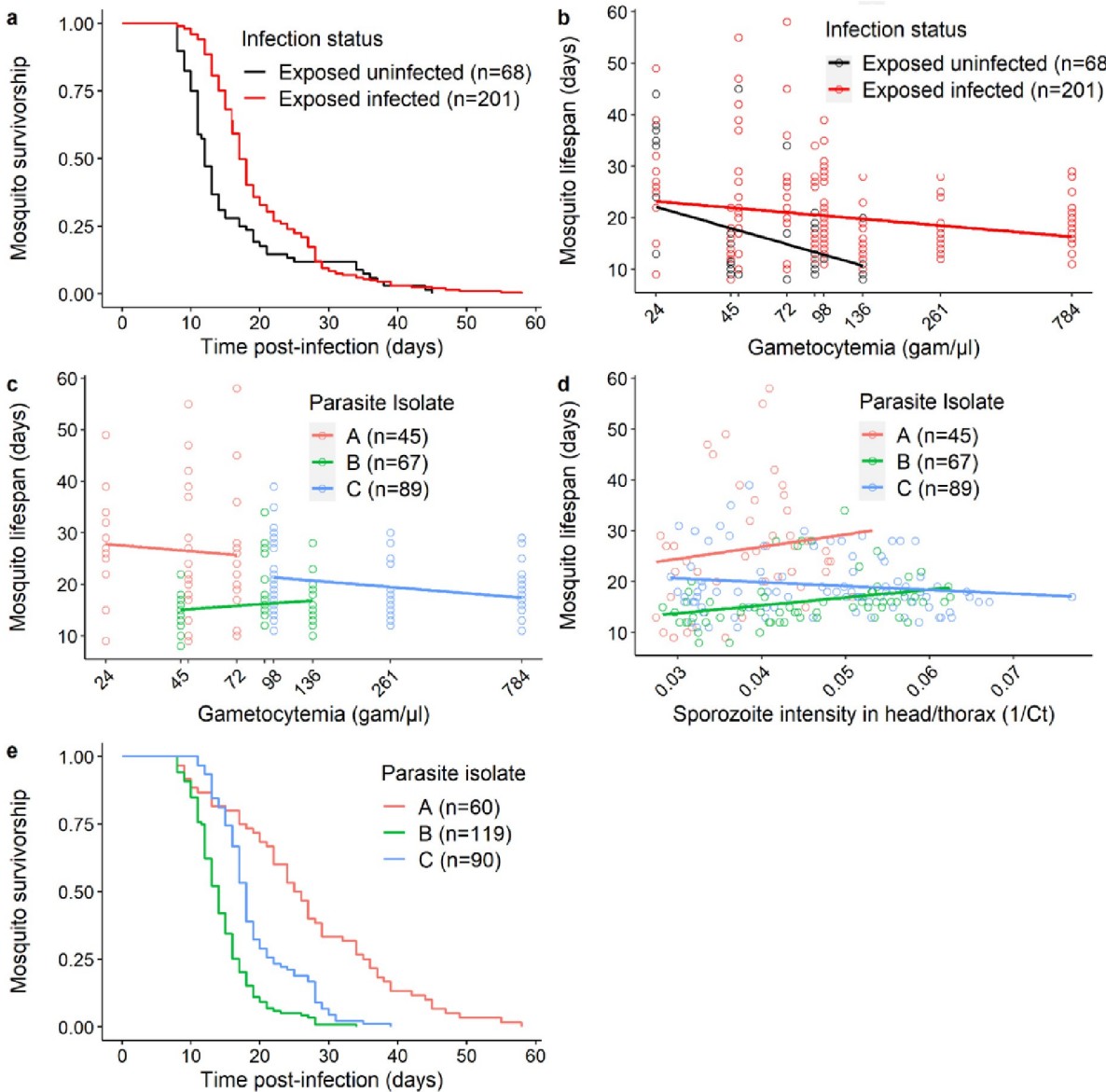

**Fig 3. Relationship between parasite density and mosquito longevity.** (a) Kaplan–Meier curves representing survival in days post blood meal for each infection status (red: mosquitoes exposed to infectious blood and having developed *Plasmodium*, black: mosquitoes exposed to infectious blood and which remained uninfected). (b) Mosquito longevity in days as a function of gametocytemia (the number of gametocytes per microliter of blood in each of the dilution groups: 24, 45, 48, 72, 91, 98, 136, 261 and 784 gametocytes/µl of blood, note that to avoid overlapping of the x-axis label, the concentrations of 48 and 91 gam/µl are not indicated). Each colored circle represents a mosquito exposed to the infectious blood (red: infected mosquitoes, black: mosquitoes that remained uninfected). The x-axis is on a log10 scale. (c) Mosquito longevity in days as a function of gametocytemia (the number of gametocytes per microliter of blood in each of the dilution groups: 24, 45, 48, 72, 91, 98, 136, 261 and 784 gametocytes/µl of blood, note that to avoid overlapping of the x-axis label, the concentrations of 48 and 91 gam/µl are not indicated). Each colored circle represents a mosquito exposed to one of the three parasite isolates (red: parasite isolate A with an initial gametocytemia of 72, green: parasite isolate B with initial gametocytemia of 136, and blue: parasite isolate C with an initial gametocytemia of 784). The x-axis is on a log10 scale. (d) mosquito longevity in days as a function of 1/Ct in mosquito infected heads/thoraxes extracts (the higher the 1/Ct value, the higher the infection intensity). For each cotton ball, 1/Ct value is the average over 3 technical replicates. Each colored circle represents an infected mosquito. The colored lines in panels (b) and (c) represent the linear relationship for each parasite isolate. (e) Kaplan–Meier curves representing survival in days post blood meal for each parasite isolate.

or thoraxes but that never produced a positive cotton (99 individuals out of 201) suggests that the inclusion of 4 to 6 technical replicates may have overestimated the proportion of positive cases by favoring false positives and support the second hypothesis. Moreover, females exposed to infectious blood but that did not develop an infection survived less well than their infected counterparts (Fig 3a), therefore providing support for the third hypothesis. As our study focused on life history traits of mosquitoes from which *P. falciparum* were detected in expelled saliva, the discrepancies between the ratio of mosquitoes positive for infection in the head/thorax compared to those positive for oocysts does not affect our conclusions.

Until recently, studying life history traits of infected mosquitoes and in particular following the dynamics of parasite development in mosquitoes required dissecting a large number of individuals and did not allow all variables to be recorded from a given female. Here, we took advantage of the recent development of a non-destructive technique to measure the presence of sporozoites in the saliva of mosquitoes as a proxy of their infectivity without sacrificing them [56]. This allows more parameters to be measured from the same individual, increasing the ability to detect trade-offs. Our results show that the manipulation of gametocyte density and subsequently of oocyst and sporozoite densities has no effect on the parasite's extrinsic incubation period. Although not significant, mosquitoes exposed to the highest gametocyte densities and carrying the highest sporozoite loads had slightly shorter EIP. This could be due to a previously identified bias in our non-destructive method of sporozoite detection. Indeed, this technique, based on the detection of sporozoite DNA in absorbent cotton on which females have come to take a sugar meal and have left saliva infected with *P. falciparum*, is subject to limitations related to detection thresholds. Consistent with this is the fact that we observed a high proportion of mosquitoes positive for parasites in head-thorax by qPCR but never produced a positive cotton ball (99 out of 201) and the fact that only a minority of cottons are positive post EIP (896 out of 2650) This suggests that, in addition to some potential false-positives among head-thoraxes, qPCR is sensitive enough to detect thousands of sporozoites within a mosquito, but reaches the limit of its sensitivity in cotton balls where sporozoites are much less numerous [56]. A consequence is that higher sporozoite loads are more likely to be detected, so our technique may overestimate the extrinsic incubation time in mosquitoes carrying low sporozoite loads [56]. Thus, despite the observed trend, which remains non-significant and likely to be due to the expected technical bias, our results highlight that the EIP does not depend on the parasite density in the system tested here. This is not consistent with the prediction that lower intensities of infection in mosquitoes would limit competition for resources between parasites and therefore speed up their development (i.e. shorter EIP). Recent results obtained by using an inbred parasite strain supported this hypothesis [43], but our results suggest a more complex relationship where, for instance, different parasite genotypes or multiplicity of infection could induce variable effects on EIP.

Regarding the survival of infected mosquitoes, the expectation based on a density-dependent cost of infection observed in other *Plasmodium*-mosquito species combinations, was that higher parasite loads would reduce the longevity of females compared to females infected with lower loads [24,44,52,54,70]. If the question of the effect of *Plasmodium* infection on mosquito survival is under debate for decades because of variance due to a large diversity of biological systems and laboratory conditions between studies, a general trend shows that infection affects survival more clearly in artificial mosquito-parasite combinations and when additional stresses occur for natural combinations of species [44,46,48,51]. In contrast, the dose-dependent nature of these effects was poorly documented for human malaria parasites in their natural vectors. Our observations did not show a correlation between mosquito lifespan and parasite load. Besides, it appears females exposed to infection, but which remained uninfected,

displayed reduced longevity compared to infected females. An explanation for this result could be a confounding effect of mosquito's size which can be correlated to longevity [71,72]. Indeed, it can be expected that larger individuals will not only ingest bigger blood meals and more infectious parasites but will also be the ones with longer lifespans. However, the expected relationships between size, longevity and infectivity are not always observed [49,51,73]. Our observation may suggest either a mutualist interaction between *P. falciparum* and *An. gambiae*, with a benefit of being infected for the host, or a cost of resistance reducing the fitness of resistant hosts. Our present data do not allow us to discriminate between the two hypotheses as a non-exposed mosquito control group would have been needed to determine if infection increases/ maintains the host's lifespan or resistance a reduced one. However, to date several studies suggest that resisting infection does induce reduced survival for mosquitoes, consistent with a cost of resistance, although considerable variation was found between assays [40,49,74]. In addition, as *Plasmodium* transmits horizontally, its effect on mosquito fecundity has only an indirect impact on its transmission, meaning it could be an adaptive strategy of the parasite to increase mosquito survival through a trade-off in energy allocation between reproduction and survival [45,75,76]. Substantial effort has been invested to decipher the trade-offs between infection, survival and fecundity for malaria vectors, but results remain controversial [19,40,77], probably because of technical difficulties to follow each of these traits for the same individual. Therefore, the non-destructive detection of parasite at the level of individual mosquitoes should allow us to better understand the possible associations between vector survival and fecundity and those between parasite load and EIP to better depict the interactions between malaria vectors and their parasites.

Regardless of the intensity of infection, our study reveals that EIP and survival varied greatly depending on parasite isolates and assay replicates. The fact that mosquitoes were exposed to the three isolates on three different days could induce confounding effects due to variation among mosquito batches. However standardized procedures in the insectary were used, including the use of a single serum sample for serum replacement during blood feeding for the entire experiment. This should have reduced batch effects, and the observed effect of parasite isolate is consistent with previous studies in which different parasites isolates were used simultanously [78,79]. In regions with high malaria transmission such as Burkina Faso, previous studies have found high genetic diversity in *P. falciparum* isolates [80,81]. Our results suggest there could be variation in the EIP and mosquito survival depending on the genetic makeup of the parasite isolates. It can be hypothesized that isolates with multiple genotypes would favor competition among genotypes and possibly for faster sporogony and more virulence [82–84]. Studies are ongoing to determine the effect of genetic diversity of parasite isolates on EIP and mosquito survival.

Our study provides evidence for the effects of *Plasmodium* parasite load in mosquito vectors on the life history traits of the mosquito and the parasite that could influence transmission. In this context, it sheds light on the potential consequences of transmission blocking interventions against malaria, which, if they do not always succeed in completely blocking the transmission of the parasite, could result in a decrease of the parasite load in mosquitoes. If a decrease in parasite load in the mosquito resulted in a strong decrease in EIP for the parasite or an increase in the longevity of vectors, the consequences in terms of transmission could be counterproductive, with an increase in the risk of human exposure to infectious bites. Our results do not show such consequences and therefore do not identify a risk associated with the decrease of parasite load in mosquitoes on the parasite's extrinsic incubation time in the mosquito or on mosquito survival, thus supporting strategies for blocking the transmission of malaria.

## Materials and methods

### Mosquitoes

In this study, we used an outbred colony of *An. gambiae* that was established in 2008 and repeatedly replenished with F1 from wild-caught female mosquitoes collected in Soumousso, (11˚23'14"N, 4˚24'42"W), 40 km from Bobo Dioulasso, south-western Burkina Faso (West Africa). To do so, field collected fed or gravid *Anopheles* females, morphologically identified as belonging to the *An. gambiae* complex, were further identified by using a SINE-PCR [85] before pooling the eggs of *An. gambiae s.s.*. Mosquitoes were then held in 30 × 30 × 30 cm mesh-covered cages and maintained under standard insectary conditions (27 ± 2 ˚C, 70 ± 5% HR, 12:12 LD) in the IRSS (Institut de Recherche en Sciences de la Santé) laboratory in Bobo Dioulasso.

### *P. falciparum* natural isolates, infectious gametocytes dilution and mosquito infection

*An. gambiae* female mosquitoes were exposed to blood samples from donors naturally infected with *P. falciparum* gametocytes using a direct membrane feeding assay (DMFA) as described previously [86] and with a dilution procedure [67].

Briefly, thick blood smears were carried out from volunteers among 5–12 year-old school-children in villages around Bobo-Dioulasso, air-dried, Giemsa-stained, and examined microscopically for the presence of *P. falciparum*. Asexual trophozoite parasite stages were counted against 200 leukocytes, while mature gametocyte stages were counted against 1,000 leukocytes and parasite densities were estimated on the basis of an average of 8,000 leuko-cytes/μl. Children with an asexual parasitaemia of > 1,000 parasites per microliter were treated according to national guidelines. Blood samples of three asymptomatic *P. falciparum* gametocyte carriers (called isolates A, B and C) were collected by venipuncture in heparin-ized tubes and their plasma was replaced by AB serum from a European donor. These blood samples underwent a series of dilutions. Dilutions involved heating part of each blood sam-ple at 45˚C for 20 minutes to inactivate the infectivity of gametocytes [73] and using this non-infectious blood to reduce the density of infectious parasites for each isolate. Isolate A, with 72 gametocytes/μl of blood, was treated to obtain three dilution factors, namely 1/1 (undiluted blood, 72 gametocytes/μl), 2/3 (48 gametocytes/μl) and 1/3 (24 gametocytes/μl). Isolate B with 136 gametocytes/μl of blood was diluted according to the same dilution factors as isolate A: 1/1 (undiluted blood, 136 gametocytes/μl), 2/3 (91 gametocytes/μl) and 1/3 (45 gametocytes/μl). The isolate C with 784 gametocytes/μl in blood was treated to obtain the dilution factor 1/1 (undiluted blood, 784 gametocytes/μl), 1/3 (261 gametocytes/μl) and 1/8 (98 gametocytes/μl).

The reconstituted blood samples were provided in feeders for one hour to female mosqui-toes aged three to six days old, distributed in 500 ml paper cups at a density of 80 mosquitoes per cup, previously starved for 12 hours. Two paper cups of 80 female mosquitoes were fed using two different feeders for each blood dilution group. After exposure to a blood meal, the unfed or partially fed females were removed and discarded. The remaining fully engorged mosquitoes were placed in 30 × 30 × 30 cm mesh-covered cages by each experimental group and kept in a bio secure room, with restricted access and cold airlock, under standard condi-tions (27 ± 2˚C, 70 ± 5% RH, 12:12 LD). The mosquitoes were given a 10% glucose solution on cotton wool after the blood meal. Mosquitoes were cold-anaesthetized for manipulation and counted at each step to verify that no accidental releases occurred. Mosquito feeding sessions were conducted three times, each time using a different parasite isolate.

## Mosquito midgut dissection

On the seventh day post blood meal (dpbm), 30 females exposed to each dilution factor of isolate A, about 24 (+/- 1) females exposed to each dilution factor of isolate B and about 10 (+/- 2) females exposed to each dilution factor of isolate C were dissected. Midguts were stained in a 1% mercurochrome solution and observed by microscopy to estimate the prevalence and intensity of oocysts in each group of exposed mosquitoes.

## Mosquito saliva collection and parasite DNA detection

A recently developed non-destructive sugar-feeding assay for parasite detection and estimating the extrinsic incubation period of *P. falciparum* in individual mosquito vectors was used [56]. Briefly, on the seventh dpbm, 20 to 40 females (median number = 30) exposed to each parasite isolate (A to C) and all experimental groups (dilution factors 1/1, 2/3, 1/3, 1/8) were individually placed in 28 ml plastic Drosophila tubes with a cotton ball (15 mg/piece) soaked with 10% glucose solution placed on each tube gauze. Cotton balls were placed at 17:00 hrs on the tubes and removed the day after at 7:00 hrs. New cotton balls were placed daily on the mosquito tubes from 8 to 22 dpbm, then at 24 dpbm and finally every four days until the mosquito died. When removed, cotton balls were stored in sterile 1.5 ml Eppendorf tubes at -20 °C for further processing.

Upon the death of all females used for saliva collection, DNA was extracted from the head and thorax of each female using the DNeasy Blood and Tissue Kit system (Qiagen, Manchester, UK) according to the manufacturer's instructions and parasite detection was carried out by qPCR, using *P. falciparum* mitochondrial DNA specific primers: qPCR-PfF 5'-TTA CAT CAG GAA TGT TTT GC-3' and qPCR-PfR 5'-ATA TTG GGA TCT CCT GCA AAT-3' [87]. For all females found positive by qPCR for *P. falciparum* in head-thorax extracts, genomic DNA from saliva in the cotton samples was also extracted using the same Qiagen protocol and the presence of sporozoites tested by the same qPCR protocol.

The DNA extracts from the heads-thoraxes were tested 4 to 6 times each for the presence of parasite DNA by two different qPCR machines and the DNA extracts from cotton were run 3 times each by the same qPCR machine. Samples were considered positive for *P. falciparum* when at least one qPCR yielded a $Ct <= 38$ and $75 <= Tm <= 80$.

## Trait measurements

**Oocyst prevalence and intensity at 7 dpbm.** For all experimental groups and for the three parasites isolates, 10 to 30 females were dissected for microscopic estimation of oocyst prevalence and intensity. Oocyst prevalence is the ratio of the number of mosquitoes with at least one oocyst to the number of all individuals dissected for each experimental group and each isolate. Oocyst intensity is the average number of oocysts in infected females for each experimental group and each parasite isolate.

**Sporozoite prevalence and intensity, extrinsic incubation period (EIP) and survival.** The females placed in individual tubes to collect saliva from cotton balls were used to analyze the prevalence and intensity of sporozoites in carcasses (heads/thoraxes) and in saliva, to measure the EIP of parasites and mosquito survival.

Sporozoite prevalence was expressed as the number of mosquito head/thoraxes detected positive for *P. falciparum* by qPCR out of the total number of dissected head/thoraxes for each treatment group and for each parasite isolate. Sporozoite intensity was expressed as the inverse of the mean number of threshold cycle during qPCR (the higher the 1/Ct value, the higher the infection intensity) for each treatment group and for each parasite isolate.

EIP was defined as the time between the infectious blood meal and the first day of positive molecular detection by qPCR of *P. falciparum* from the cotton wool where the female deposited saliva during sugar feeding.

Dead mosquitoes in the individual tubes of each experimental group were recorded every morning at 8:00 hrs to record mosquito survival in each experimental group.

## Statistical analyses

All statistical analyses were performed by R (version 4.0.2). The effect of gametocyte density on oocyst and sporozoite prevalence was tested using logistic regression for generalized linear mixed models (GLMM, binomial errors, logit link; "lme4 package"), and its effect on oocyst and sporozoite density was tested using a negative binomial GLMM and a linear mixed model ("lme4" package), respectively. In these models, gametocyte density was set as both a fixed and a random slope effect and parasite isolate as a random intercept. EIP was analysed using two LMMs, the first specifying gametocyte density as a fixed and a random slope factor and parasite isolate as a random intercept, the second specifying sporozoite load in mosquito head/thorax as a fixed and a random slope factor and parasite isolate as a random intercept. We also investigated the effect of parasite isolate (set as a fixed effect) on EIP using a Cox's proportional hazards regression model. The effect of infection status (infected vs. uninfected) on mosquito survival was evaluated using a mixed Cox's proportional hazards regression model (package "coxme") with infection considered as a fixed effect and parasite isolate as a random intercept effect. Mosquito longevity was also analysed using two LMMs, the first specifying gametocyte density as a fixed and a random slope factor and parasite isolate as a random intercept, the second specifying sporozoite load in mosquito head/thorax as a fixed and a random slope factor and parasite isolate as a random intercept. Finally, we investigated the effect of parasite isolate (set as a fixed effect) on mosquito survival using a Cox's proportional hazards regression model. For each model, the statistical significance of the fixed effects was evaluated using the "Anova" function of the "car" package.

## Acknowledgments

We thank all volunteers for participating in this study as well as the local authorities for their support. We are very grateful to all the students and technicians at the IRSS/IRD who provided valuable assistance for the experiments in this study and to Philip Agnew for correcting our English.

## Author Contributions

**Conceptualization:** Edwige Guissou, Thierry Lefèvre, Anna Cohuet.

**Data curation:** Edwige Guissou, Thierry Lefèvre, Anna Cohuet.

**Formal analysis:** Edwige Guissou, Thierry Lefèvre, Anna Cohuet.

**Funding acquisition:** Domombabele François de Sales Hien, Thierry Lefèvre, Anna Cohuet.

**Methodology:** Edwige Guissou, Koudraogo Bienvenue Yameogo, Thierry Lefèvre, Anna Cohuet.

**Supervision:** Dari Frédéric Da, Domombabele François de Sales Hien, Serge Rakiswende Yerbanga, Georges Anicet Ouédraogo, Kounbobr Roch Dabiré, Thierry Lefèvre, Anna Cohuet.

**Validation:** Thierry Lefèvre, Anna Cohuet.

**Visualization:** Edwige Guissou, Thierry Lefèvre, Anna Cohuet.

**Writing – original draft:** Edwige Guissou, Dari Frédéric Da, Thierry Lefèvre, Anna Cohuet.

**Writing – review & editing:** Edwige Guissou, Thierry Lefèvre, Anna Cohuet.

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
