## [Decision Letter · Decision Letter 0]

25 Feb 2023

Dear Mrs Guissou,

Thank you very much for submitting your manuscript "Intervention reducing malaria parasite load in vector mosquitoes: no impact on Plasmodium falciparum extrinsic incubation period and the survival of Anopheles gambiae" for consideration at PLOS Pathogens. As with all papers reviewed by the journal, your manuscript was reviewed by members of the editorial board and by several independent reviewers. The reviewers appreciated the attention to an important topic. Based on the reviews, we are likely to accept this manuscript for publication, providing that you modify the manuscript according to the review recommendations.

Sincerely,

Kenneth D Vernick

Academic Editor

PLOS Pathogens

Ronald Swanstrom

Section Editor

PLOS Pathogens

Kasturi Haldar

Editor-in-Chief

PLOS Pathogens

orcid.org/0000-0001-5065-158X

Michael Malim

Editor-in-Chief

PLOS Pathogens

orcid.org/0000-0002-7699-2064

Reviewer Comments (if any, and for reference):

Reviewer's Responses to Questions

**Part I - Summary**

Reviewer #1: This study used a recently developed non-destructive feeding assay in An. gambiae to test for relationships between gametocyte load in the blood and parasite DNA in saliva (over a time course) and in tissues at the end of the collection period. In females with positive saliva samples, EIP was estimated. No relationship was found between gametocytaemia and EIP, nor between tissue parasite loads and EIP, but parasite isolate was a significant factor. Parasite infection generally improved mosquito survival, and this effect varied as above by parasite isolate. The work is novel in using the assay to examine traditionally difficult-to-assess relationships. The paper is framed around implications for TBI – although I actually think the most interesting aspects are really about basic parasite & vector interactions. Most interesting is the evidence that parasite genotype is the biggest driver of EIP. This will surely lead to future studies.

Strengths of the study include mosquitoes were fed on naturally infected patient bloods. Good experimental design, replication, and statistical analysis. Novel approach to address some long standing questions.

Weaknesses include some issues with clarity and presentation.

Reviewer #2: This is an important study to answer key question for TBI. The authors investigated the impact of parasite load on key parameters: oocyst prevalence, intensity, EIP and mosquito survival. The parasite strain however, did impact on some of there parameters. The authors should discuss if these outcomes might be similar in other vector species or even in other parasite strains from e.g. East Africa

Reviewer #3: This paper describes results of experiments designed to assess how variation in the gametocyte density of human blood infected with P. falciparum impacts the subsequent development time of parasites in mosquitoes (EIP) and their survival. The motivation is described as understanding the potential for ‘unexpected negative consequences’ of transmission-blocking interventions (TBIs), which would work by reducing but not completing blocking parasite transmission to mosquitoes. Strengths are the use of a real human malaria parasite system; including parasite isolates from natural infections and an outbred mosquito vector population. Other strengths are the use of a novel method for manipulating the density of parasites in the infectious blood meal while standardizing other components; and a novel non-invasive method for estimating when mosquitoes first become invasive. The resultant data would be of general interest and value to malaria-vector interactions.

However, there are some limitations and weaknesses that I recommend be addressed before this is published. In particular, engagement with the existing, quite large body of literature in this area is very superficial; and falls short in terms of contextualizing findings in the bigger picture. I have summarized key points below.

More specific comments and suggestions are provided in the uploaded file.

Reviewer #4: In this study, Guissou and colleagues took advantage of a non-destructive technique that they previously developed (Guissou et al., 2021) to assess the effect of different gametocyte/oocyst loads on the extrinsic incubation period (EIP) of the parasite and mosquito survival. Using P. falciparum field isolates, diluting infected gametocyte blood to different densities had no effect on the prevalence of infection at 7dpi. As expected, oocyst intensity decreased with increasing dilution. Consistently, the proportion of sporozoite-positive mosquitoes, assessed by qPCR on parasite DNA extracted from thoraces and head at the time of death, significantly increased with the density of infectious gametocytes. Infection intensity/gametocytemia had no impact on EIP, suggesting no risk associated with reduced parasite load after a TBI. Interestingly, infected mosquitoes lived longer than non-infected counterparts.

The study is thorough, and the results are interesting and well discussed. There are a couple of issues I found when reading the manuscript.

Reviewer #5: In this paper, the authors investigate whether decreasing the malaria parasite load within vector mosquitoes might have unintended consequences. Specifically, whether less heavily infected mosquitoes might live longer or if this might lead to parasites progressing through their life cycle more quickly (a shorter incubation period within the mosquitoes). The authors rightly point out that these factors must be properly considered before the impact of transmission-blocking interventions can be fully appreciated. A number of these tools are passing through the development pathways, which makes this work very timely. The variation in results seen for different parasite isolates is quite high, which makes it more difficult to extrapolate the implications of the findings more widely. Nevertheless, the use of natural isolates of P. falciparum is also one of the strengths of the work. The work is well presented, and I think it will be of interest to the malaria research community. I have a few comments and queries, however, that should be addressed before the article can be approved for publication (see attached document). I commend the authors for this interesting work.

**Part II – Major Issues: Key Experiments Required for Acceptance**

Reviewer #1: None

Reviewer #2: None

Reviewer #3: I am not recommending any further experiments as being required. One potentially useful, additional statistical analysis could be to assess the relationship between parasite density as assessed via qPCR in head/thorax samples and the parasite positivity of the 'cotton ball' samples that the mosquitoes salivated on. A high number of qPCR positive mosquitoes had 'negative' cotton ball samples. It would be useful to know whether was influenced by parasite density (e.g were mosquitoes with high infection loads via qPCR more likely to generate positive cotton ball samples).

Reviewer #4: As a general comment, the figures are difficult to read. The authors should provide a more detailed legend with a description of the results and stats.

Moreover, also the result section doesn’t provide sufficient experimental details, so that drawing conclusions is not always possible. For instance, were all infection experiments done in parallel (all 3 gam carriers and all dilutions) or were they done using mosquitoes from different days (for instance, one gam carrier/day)? If the latter, this may explain the results of mosquito longevity after infection with the different carriers, which are otherwise frankly difficult to explain. Indeed, different mosquito batches often have different survival. Different replacement serum could also be a cause of mortality. While the intra-gam carrier comparison using the 3 dilution is solid, an intra gam carrier comparison carried out over different days would not be appropriate, and therefore should be removed from the manuscript as not biologically relevant. Lines 283-291 do not sufficiently discuss to discuss this issue. Same considerations for the EIP comparison between the three gametocyte carriers, and those data should also be removed or more amply discussed.

The discrepancy between oocyst prevalence data and sporozoite prevalence data is puzzling. The authors do a good job at identifying possible reasons behind this discrepancy, but small size of oocysts is not likely to be a factor given most other labs (and their lab in previous publications) have good correlation between oocyst and sporozoite data. The authors should repeat the statistical analysis after removing those mosquitoes that did not show sporozoite in saliva at any time. This way if the discrepancy is due to false positives detected by qPCR, they’d obtain more reliable results. Both data sets could be discussed and compared, which would also be useful for other studies that are looking into qPCR for sporozoite detection.

Reviewer #5: (No Response)

**Part III – Minor Issues: Editorial and Data Presentation Modifications**

Reviewer #1: I have some suggestions for the introduction. It assumes some knowledge that the readers may not have if this is to be of broad interest. Please add some examples of types of TBIs. Also, explain how TRA is measured and what it is in practice. Also, the third paragraph sets up the entire argument for why this study should be done. It is a jumble of ideas and some sentences are not clear. Line 102 What is a thinner explanation? Awkward language. Lines 1-6-110. Not clear what you mean and this is the premise for the paper. Lines 129-130 In consequences to?? I think this entire paragraph needs attention.

Methods

What is the justification for checking mosquitoes at 7dpi? Is this standard in the field, if so, why? Hard to do multiple time points given the scale, but explain this choice. Would results have differed if other time points were checked?

Results

You have room on your graphs. Is it possible to use a single descriptive word instead of A, B, C? The graphs are impenetrable without looking back and forth many times to the figure legend. Since A, B, and C can often be very different things – I think important to highlight that visually immediately on the graph.

Also, A, B, C are used to mean different variables in first sets of graphs and then isolates in others. Use numbers instead when they are meant to indicate isolates.

Reviewer #2: The reference provided to report on the extend of insecticide resistance is outdated and will be good to include more up to date citations to reflect the extend of the problem in more recent times.

Reviewer #3: There are quite a number of small grammar issues throughout (e.g LN 73 – should be ‘spraying’ not ‘sprayings’. LN 79 – should be “parasites” not ‘parasite’). I recommend the authors do a more thorough proof reading for the revised version.

Reviewer #4: Line 80: there are also other strategies that use drugs directly in the mosquito (PMID: 30814727)

Line 129: ref PMID: 12036738 (cited by the authors but in a different context) shows that P. falciparum does not affect mosquito survival, so the sentence stating that infection intensity is reported to negatively impact survival isn’t correct

Reviewer #5: (No Response)

PLOS authors have the option to publish the peer review history of their article (what does this mean?). If published, this will include your full peer review and any attached files.

Reviewer #1: No

Reviewer #2: No

Reviewer #3: No

Reviewer #4: No

Reviewer #5: No

Figure Files:

Data Requirements:

Reproducibility:

References:

---

## [Editor Report · Decision Letter 1]

18 Apr 2023

Dear Mrs Guissou,

We are pleased to inform you that your manuscript 'Intervention reducing malaria parasite load in vector mosquitoes: no impact on Plasmodium falciparum extrinsic incubation period and the survival of Anopheles gambiae' has been provisionally accepted for publication in PLOS Pathogens.

Best regards,

Kenneth D Vernick

Academic Editor

PLOS Pathogens

Ronald Swanstrom

Section Editor

PLOS Pathogens

Kasturi Haldar

Editor-in-Chief

PLOS Pathogens

orcid.org/0000-0001-5065-158X

Michael Malim

Editor-in-Chief

PLOS Pathogens

orcid.org/0000-0002-7699-2064
---

## [Editor Report · Acceptance letter]

4 May 2023

Dear Mrs Guissou,

We are delighted to inform you that your manuscript, "Intervention reducing malaria parasite load in vector mosquitoes: no impact on *Plasmodium falciparum* extrinsic incubation period and the survival of *Anopheles gambiae*," has been formally accepted for publication in PLOS Pathogens.

Best regards,

Kasturi Haldar

Editor-in-Chief

PLOS Pathogens

orcid.org/0000-0001-5065-158X

Michael Malim

Editor-in-Chief

PLOS Pathogens

orcid.org/0000-0002-7699-2064